# Scaled Sea Surface Design and RCS Measurement Based on Rough Film Medium

**DOI:** 10.3390/s22166290

**Published:** 2022-08-21

**Authors:** Chenyu Guo, Hongxia Ye, Yi Zhou, Yonggang Xu, Longxiang Wang

**Affiliations:** 1Key Laboratory for Information Sciences of Electromagnetic Waves, Fudan University, Shanghai 200433, China; 2Science and Technology on Electromagnetic Scattering Laboratory, Shanghai 200438, China

**Keywords:** scaled measurement, rough sea surface scattering, equivalent permittivity

## Abstract

The electromagnetic (EM) scattering characteristics of the rough sea surface is very important for target surveying and detection in a sea environment. This work proposes a scaled sea surface designing method based on a rough thin-film medium. For the prototype sea surface, the permittivity is calculated with the seawater temperature, salinity, and EM wave frequency according to the Debye model. The scale film material is mixed with carbon black and epoxy, whose volume ratio is optimized with the genetic algorithm through the existing electromagnetic parameter library. This method can overcome the previous difficulties of adjusting the same permittivity of the prototype sea water. According to the EM scaled theory, the scaled geometric sample is numerically generated with the D-V spectrum for the given wind speed, and is fabricated using 3D printing to keep the similar seawater shape. Then, the sample is sprayed with a layer of film material for EM scattering measurement. The simulated and measured radar cross-section (RCS) results show good consistency for the prototype seawater and scaled materials, which indicates the proposed scaled method is a more efficient method to get the seawater scattering characteristics.

## 1. Introduction

The electromagnetic (EM) scattering characteristics of the complex natural sea surface are very important for integrated perception information of marine ecological environments, ocean battle scenes, and so on [1]. When satellite or airborne radars detect and warn targets (such as ships, aircraft, etc.) in the actual sea environment, the echo signal will inevitably include clutter of the rough sea surface. Ocean environment information such as a wind field, sea surface salinity (SSS), and sea surface temperature (SST) can be obtained from the observation data of meteorological or ocean satellites according to the EM scattering mechanism of the sea surface [2,3,4]. The theoretical research on EM scattering of the rough sea surface mainly focuses on the calculation of rough surface scattering, such as forward and backward method (FBM), sparse matrix canonical grid (SMCG), finite element domain decomposition method (FEDDM), etc. [5,6,7,8]. However, numerical calculation methods are often limited by the performance of computer hardware, which limits their simulation scale. In addition, numerical methods are not able to consider the factors of the detection system. Therefore, it is necessary to develop rough sea surface measurement technology.

Radar cross-section (RCS) is an important parameter describing the EM scattering characteristics, which is defined as 4π times of the ratio of the scattered power received by the target in a unit solid angle to the power density of the incident wave on the target [9]. The measurement of RCS often requires a large enough area to carry out measurement experiments, especially for scatter measurements involving complex marine environments, and often consumes huge resources and manpower. In addition, due to the uncertainty of the actual measurement environment, it is impossible to achieve a deterministic analysis of the detection environment. With the development of microwave devices and measurement technology, the laboratory measurements of RCS have become possible [10,11]. The measurement method in laboratories is mainly divided into the far-field method, the compact field method, and the near-field method. The far-field method often needs a very large area so that the measured target is far enough from the measuring antenna. The compact field method is too difficult to detect in a complex ocean environment, which makes it difficult to operate. The near-field measurements often use a limited beam to illuminate the target. The illumination of a narrow beam on the sea surface often causes false edge diffraction from the sea surface edge. In view of the above difficulties, the scaled measurement uses the scaled model of the prototype surface, and infers the measured data to the RCS of the prototype surface based on the EM scaled theory.

The scaled model design often involves metallic and non-metallic objects. According to the previous scale method, the scaled materials should have the same permittivity as the prototype ones [5,12]. Since the seawater is a dispersive medium, the magnetic loss coating on the metallic surface [13,14] is designed according to the reflection loss of the coating backed by a perfect conductive plate. Reference [5] presented a scale method to simulate the dynamic sea surface in a water basin. This method requires that the water salinity and temperature must be changed simultaneously to get the appropriate permittivity of the prototype seawater. It is still limited for the permittivity value. As the scale frequency increases to hundreds of GHz, it is nearly impossible to achieve according to the relationship of permittivity with salinity or temperature [15]. Moreover, the salt will corrode the equipment in the water basin, so it is a destructive measurement. Therefore, how to support a new way to establish the practical permittivity and a low-cost scaled seawater material is still valuable for designing the scaled sea environment. With the proper scaled seawater material, the complex scattering characteristics of the rough sea surface could be easily obtained in a laboratory.

This article is arranged as follows. Section 2 deduces the relationship of scaled model and the prototype one according to the EM scaling principle. Section 3 introduces the design and construction process of the scaled sea surface model. Section 4 gives the RCS results of the prototype sea surface with the simulation and measurement methods, as well as the performance analysis of the rough film medium for a scale rough sea surface. Finally, the conclusion is presented in Section 5.

## 2. Scaling Principle for Rough Sea Surface Scattering

### 2.1. The Scaling Principle for Scaled Sea Surface Using Rough Thin-Film Media

The scaling principle for a sea surface with a rough film medium is mainly to get the same reflection and scattering of electromagnetic waves as the prototype sea surface. The calculation of electromagnetic scattering from the sea surface mainly includes high-frequency approximate calculation and precise calculation by numerical methods [6]. According to the similarity criterion, the geometric size of the prototype sea surface and the scale model must be in good proportionality. Moreover, the permittivity in two electromagnetic systems must remain strictly the same. When the prototype material is a dispersive material, it is difficult to construct the scaled materials which have the same dispersion relationship as that of the prototype material. For the structure of the scaled sea surface, we use the principle of the high-frequency approximation algorithm to optimize the electromagnetic parameters of the scaled material. The starting point of the physical optics method is derived from the Stratton-Chu scattering field integral equation, in which the total scattering electric field and the scattering magnetic field formula on the surface of the object are expressed as follows [7].
(1)$${\bar{E}}_{s}\left(\bar{r}\right)=\frac{ike^{ikr}}{4\pi r}\iint \left[Z{\bar{J}}_{s}\left({\bar{r}}^{\prime }\right)+\hat{s}\times {\bar{M}}_{s}\left({\bar{r}}^{\prime }\right)\right]e^{-i{\bar{k}}_{s}\cdot {\bar{r}}^{\prime }}ds^{\prime }$$Here, k is the propagation constant of free space, Z is the intrinsic impedance of free space, $\hat{s}$ is the unit vector of the scattered wave, $\bar{r}$ is the position vector of the point on the surface, $r=\left|\bar{r}\right|$. ${\bar{J}}_{s}=\hat{n}\times \bar{H}$ is the surface current, ${\bar{M}}_{s}=\hat{n}\times \bar{E}$ is the surface magnetic current, $\hat{n}$ is the normal unit vector outside the surface, and $\bar{H}$ and $\bar{E}$ are the total magnetic and electric field on the rough sea surface, respectively.

According to the Snell reflection laws of a flat dielectric interface, the induced current on the flat interface can be expressed as
(2)$$Z{\bar{J}}_{s}=\left({\bar{E}}_{i}\cdot {\hat{p}}_{i}\right)\left(\hat{n}\times {\hat{q}}_{i}\right)\left(1+R_{v}\right)-\left({\bar{E}}_{i}\cdot {\hat{q}}_{i}\right)\left(\hat{n}\cdot {\hat{k}}_{i}\right)\left(1-R_{h}\right)$$
(3)$${\bar{M}}_{s}=\left({\bar{E}}_{i}\cdot {\hat{p}}_{i}\right)\left(\hat{n}\cdot {\hat{k}}_{i}\right)\left(1-R_{v}\right)+\left({\bar{E}}_{i}\cdot {\hat{q}}_{i}\right)\left(\hat{n}\times {\hat{q}}_{i}\right)\left(1+R_{h}\right)$$Here, ${\bar{E}}_{i}$ is the electric field vector of the incident plane wave.$\;{\hat{q}}_{i}=\frac{{\hat{k}}_{i}\times \hat{n}}{\left|{\hat{k}}_{i}\times \hat{n}\right|}$, ${\hat{p}}_{i}={\hat{q}}_{i}\times {\hat{k}}_{i}$ are the perpendicular and parallel polarization vectors of the flat interface. Therefore, when the reflection coefficients at prototype frequency $f_{0}$ and scaled frequency $f_{1}$ are equal, i.e., $R_{v,h}\left(f_{0}\right)=R_{v,h}\left(f_{1}\right)$, the induced current ${\bar{J}}_{s}$ and ${\bar{M}}_{s}$ will be equal, and the prototype material and the scaled-down material can meet the strict similarity criterion. However, when the sea surface model is scaled down from low frequency (for example 7.75−7.85 GHz) to high frequency (23.25−23.55 GHz for a three-times reduction ratio), it is difficult to guarantee that the permittivity of the scaled rough surface is exactly equal to that of the prototype rough surface.

This paper intends to use the equivalent electromagnetic model based on the reflection coefficient approximation to design the scaled sea surface at multiple angles. Then, the RCS of the scaled surface is expressed as
(4)$$\sigma^{\prime }=\underset{r^{\prime }\to \infty }{\lim}\left({4\pi r^{\prime }}^{2}\frac{{{\bar{E}}^{\prime }}_{s}\times {{\bar{H}}^{\prime }}_{s}}{{{\bar{E}}^{\prime }}_{i}\times {{\bar{H}}^{\prime }}_{i}}\right)={\left(K_{L}\right)}^{2}\sigma$$Here, $K_{L}$ is the scaling factor. The field quantities with an apostrophe are those in the scaled system. Equation (5) is the scaling principle between the scaled system and the prototype system, which can be transformed into logarithmic form as
(5)$$10\log\sigma^{\prime }=20\log K_{L}+10\log\sigma$$

### 2.2. Reflection Coefficients of Seawater

The reflection coefficient of the sea surface can be expressed as a function of the permittivity at a given incidence angle. The reflection coefficient under vertical polarization and horizontal polarization is expressed as [16,17]
(6)$$R_{v}=\frac{\varepsilon_{r}\cos\theta_{i}-\sqrt{\varepsilon_{r}-\sin^{2}\theta_{i}}}{\varepsilon_{r}\cos\theta_{i}+\sqrt{\varepsilon_{r}-\sin^{2}\theta_{i}}},\;R_{h}=\frac{\cos\theta_{i}-\sqrt{\varepsilon_{r}-\sin^{2}\theta_{i}}}{\cos\theta_{i}+\sqrt{\varepsilon_{r}-\sin^{2}\theta_{i}}}$$Here, $\theta_{i}$ is the incident angle of electromagnetic wave and $\varepsilon_{r}$ is the complex relative permittivity of seawater. The permittivity of sea water is often a complicated function of seawater temperature T, seawater salinity S, and electromagnetic wave frequency f. The GW2020 model was developed by Yiwen Zhou et al. [18]. using the measurement data from advanced equipment. The inversion experiment of the measured data shows that this model has more advantages than the KS model [19] and the MW2012 [20] model in seawater salinity inversion. According to the GW2020 model, the relative permittivity of seawater is expressed as follows.
(7)$$\varepsilon_{r}\left(S,T\right)=4.9+\frac{\varepsilon_{s}\left(T\right)R\left(S,T\right)-4.9}{1+i2\pi f\tau \left(T\right)}-i\frac{\sigma \left(S,T\right)}{2\pi f\varepsilon_{0}}$$Here, $\varepsilon_{s}\left(T\right)$ is the permittivity of pure water at rest frequency (f=0), $\varepsilon_{0}$ is the vacuum permittivity, $\tau \left(T\right)$ is Debye relaxation time, S is salinity (psu), T is temperature (°C), and f is incident wave frequency (Hz). The specific Debye parameter forms are fitted as follows with the measured data.
(8)$$\tau \left(T\right)=1.7503e^{-11}-6.1299e^{-13}T+1.2451e^{-14}T^{2}-1.1493e^{-16}T^{3}$$
(9)$$\varepsilon_{s}\left(T\right)=8.8052e-4.0179e^{-1}T-5.1027e^{-5}T^{2}+2.5589e^{-5}T^{3}$$
(10)$$R\left(S,T\right)=1-3.9719e^{-3}S+2.4921e^{-5}ST+4.2756e^{-5}S^{2}-3.9283e^{-7}S^{2}T-4.1535e^{-7}S^{3}\;\sigma \left(S,T\right)=\left(9.5047e^{-2}\;S-4.3086e^{-4}S^{2}+2.1618e^{-6}S^{3}\right)\times [1+3.7602e^{-2}T+$$
(11)$$6.3283e^{-5}T^{2}+4.8342e^{-7}T^{3}-3.9748e^{-4}ST+6.2652e^{-6}S^{2}T]$$

Supposing sea surface temperature T=20 °C, Figure 1 shows the permittivity of pure water and seawater with the GW2020 model. Suppose sea surface temperature T=20 °C, salinity S=30 psu, frequency f=7.8 GHz, the permittivity of seawater is $\varepsilon_{r}=61.7-35.5\;i$. Figure 2 shows the reflection coefficient amplitude with the incident angle under vertical polarization and horizontal polarization. Since $R_{v}$ changes remarkably with incident angles, it is used for later parameter optimization.

## 3. Design and Manufacture of Scaled Sea Surface

In order to make the RCS of the prototype sea surface and the scaled sea surface meet the scale ratio formula, we need to spray the scaled sea surface model with a layer of equivalent material, so that the reflection coefficient of the scaled sea surface model is approximately equal to the reflection coefficient of the prototype sea surface. In this section, the equivalent material is first designed, and the scaled sea surface model is manufactured for measurement in the scale system.

### 3.1. Permittivity Measurement Based on the Waveguide Method

According to the transmission/reflection method [21], the permittivity can be reversed as
(12)εr=λ0λc2+1Z˜c21−λ0λc2Here, $\lambda_{0}$ is the length of an electromagnetic wave in air, $\lambda_{c}$ is the cutoff wavelength of the waveguide. ${\tilde{Z}}_{c}$ is expressed as
(13)Z˜c=1+Γc1+ΓcHere, $\Gamma_{c}$ is the reflection coefficient of the measured sample surface.

### 3.2. Preparation of the Surface Material for Simulating a Rough Sea Surface

Carbon black is a kind of conductive material, and its conductivity is closely related to its microstructure, particle size, and structure and surface properties. The actual material is shown in Figure 3. The density of carbon black is 2.00 g/cm^3^ and the density of paraffin is 0.90 g/cm^3^.

The crystal phase of the powder was estimated by an X-ray diffractometer (D/teX Ultra250, Rigaku Smartlab II, Tokyo, Japan) with Cu K-radiation (wavelength = 0.154 nm), and the scan step size was 0.02 deg/s with 50 steps/degree. The morphology of the composites was measured using scanning electron microscopy (German ZEISS GeminiSEM 300). The raw commercial carbon black was supplied by Jinan Hongshun Chemical Co. Ltd., Jinan, China. The morphology of the carbon black cluster and the particle dispersion in the resin can be shown in Figure 4. The aggregated nano carbon black had a spherical shape with a diameter about 20~50 nm, and the carbon black cluster was uniformly dispersed in the epoxy resin.

Figure 5 shows the XRD patterns of the carbon black particle. The diffraction peaks (0 0 2) and (1 0 0) of the carbon show typical carbonous material characteristics, while the other very low intensity value shows a low amount of disorder in the samples.

In this section, we make the scaled seawater materials by mixing a certain proportion of carbon black and paraffin. The carbon black mainly determines its dielectric properties; the paraffin is treated as a lossless connecting agent. The volume ratio of carbon black determines the permittivity of the mixed material. The preparation process is divided into the following steps:(a)Weigh two parts of paraffin with a mass of 1 g. Then, weigh the carbon black with a mass of 0.75 g and 0.95 g, which corresponds to the volume ratio of 25% or 30% in the mixed material, respectively.(b)Pour the weighed paraffin into the small beaker and heat it on the heating table. After the paraffin dissolves, the carbon black is timely poured into the small beaker and stirred. During mixing, the small beaker shall not leave the heating table to prevent the mixed materials from curing rapidly, resulting in uneven mixed materials. Then, the mixed material is solidified quickly.(c)An appropriate amount of evenly mixed material is poured into the metal groove on the heating table. The thickness of the metal groove is 2.22 mm, so the thickness of the final measured sample is also 2.22 mm. When the mixture fills the metal groove evenly, a rectangular sample is cut with a blade, as shown in Figure 6.

### 3.3. Measurement Data of Scattering Parameters and Inversion of Permittivity

Suppose that the prototype frequency is 7.75−7.85 GHz (center frequency is 7.8 GHz), and the scaled factor $K_{L}=3$, so the scaled frequency is 23.25−23.55 GHz. The waveguide system is used to measure the permittivity. As shown in Figure 7, the measuring system consists of a vector network analyzer (VNA), a coaxial line, a coaxial waveguide converter, and a BJ220 rectangular waveguide. The wave transmission mode is usually ${\mathrm{TE}}_{10}$ wave, and the frequency range is 17.6−26.7 GHz.

The sample (mixture of carbon black and paraffin) is easily placed unevenly and has a certain thickness. The non-uniformity of the contact between the waveguide surface and the sample surface will lead to a certain deviation between $S_{11}$ and $S_{22}$ and $S_{12}$ and $S_{21}$. The measurement results show that this difference is very small. In order to minimize the influence of this deviation on the inversion results, $S_{11}=S_{22}=\left(S_{11}+S_{22}\right)/2\;$and $S_{12}=S_{21}=\left(S_{12}+S_{21}\right)/2$ are taken for inversion calculation.

Figure 8 shows the measured $S_{11}/S_{22}$ parameters and $S_{21}/S_{12}$ parameters and the retrieved dielectric constant values. The vertical axis of S parameters is in dB, that is, $20\log\left|S\right|$. When the carbon black volume ratio is 25%, the complex relative permittivity of the mixed material is about 32−14i between 23.25 and 23.55 GHz. When the volume ratio is 30%, the complex relative permittivity of the mixed material is about 68−43i. Moreover, if the volume ratio is 0%, the complex relative permittivity of pure paraffin material is about 2−0.01i.

### 3.4. Optimization of Permittivity of the Rough Film

The geometric model of the scaled rough surface can be made by 3D printing. Then, it is sprayed with a layer of film material to generate the scaled rough sea surface model. According to the scaled condition in Section 2, if the reflection coefficients of the scaled surface model and the prototype sea surface model are approximately equal in all incident angles, the RCS of the corresponding rough surfaces will meet the scaling formula. For the given prototype sea surface (including the frequency and sea surface temperature and salinity), the permittivity $\varepsilon_{r}$ is calculated according to the Debye model of seawater, and the corresponding reflection coefficient $R_{\mathrm{full}}$ is also calculated for different incident angles. Then, for a given scaled coefficient $K_{L}$, the volume ratio of carbon black and paraffin for the scale model is optimized through the existing electromagnetic parameter library. The optimization function is expressed as
(14)$$\underset{}{\min}\overset{90{}^{\circ}}{\underset{\theta_{i}=0{}^{\circ}}{\sum }}\left|R_{\mathrm{full}}-R_{\mathrm{scall}}\right|$$Here, *R*_scale_ is the reflection coefficient of the scaled sea surface. Then, the best volume ratio of carbon black in the mixture is optimized with the genetic algorithm to minimize the above optimization function. The optimization process is as follows [22,23,24].

(1)Initialization of population size, crossover probability, mutation probability, and iteration number. The random function is used to generate 0 or 1, and a single population is binary coded to generate the initial population. The coding length is related to the scale of the volume ratio and its discrete precision. Here, the discrete precision is set as 1%.(2)Electromagnetic parameter fitting calculation. The spline interpolation method is used to calculate the equivalent electromagnetic parameters (complex relative dielectric constant) of each proportion of mixed scaled materials.(3)Calculate the fitness of each individual in the population according to the optimization function in Equation (9). Judge the termination conditions so that the optimization function is greater than the set parameter (it is set as 80 in this paper) according to the individual with the largest fitness in the population. If so, the calculation ends, otherwise go to the next step.(4)Copy the binary code of the old group to obtain the new group according to the crossing probability and crossing point.(5)Select appropriate chromosomes (referring to the binary code of a single individual) to participate in the crossover according to the crossover probability and crossover point. The original chromosome is replaced with the generated new chromosome.(6)Randomly select some genes to participate in mutation according to the mutation probability (that is, their binary code changes from 0 to 1 or from 1 to 0). With this operation, a new population is obtained again.(7)Steps 4–6 generate the new generation population. Calculate the fitness of the new generation population, and compare it with the previous population. If the fitness of the new population is greater than that of the previous population, go to step 3 with the new population. Otherwise, the new population will be eliminated and the previous population will be used to transfer to Step 3.

According to the above optimization function, the reflection coefficient performance parameters can be expressed as a function of four variables (material addition ratio, the real part of the complex relative dielectric constant, the imaginary part of the complex relative dielectric constant, and the incident angle of the electromagnetic wave). The decoding process is to convert the binary number of the final optimized proportion into a value between the maximum filling proportion and the minimum filling proportion.

### 3.5. Physical Preparation of the Rough Sea Surface Model

The real rough sea surface is usually considered to be composed of numerous random sine waves with different amplitudes, wave numbers, and phases. Thus, it can be described with some statistical parameters. As an important statistical characteristic, a wave spectrum is defined as the Fourier transform of the rough sea surface’s autocorrelation function, which describes the internal energy distribution of the wave with respect to frequency and direction. The geometric generation of the rough sea surface is generally realized by the wave spectrum. The Monte Carlo method is based on the assumption that the sea surface is linearly superimposed by countless sine waves. The height fluctuation of the sea surface is obtained by filtering the wave spectrum and then the inverse Fourier transform. The wave spectrum used in this paper was proposed by Durden and Vesecky based on the scattering cross-section data of RADSCAT. The form of the DV spectrum is [25].
(15)$$W\left(k,\varphi \right)=\frac{1}{2\pi k}S\left(k\right)\Phi \left(k,\varphi \right)$$
(16)$$S\left(k\right)=\frac{0.001}{\pi k^{4}}\left\{\begin{array}{c} \exp\left[0.74k_{c}^{2}\left(\frac{1}{4}-\frac{1}{k^{2}}\right)\right]\;k<2\\ {\left(\frac{bku_{*}^{2}}{g+\gamma k^{2}}\right)}^{alog\left(\frac{k}{2}\right)}\;k\geq 2 \end{array}\right.$$Here, $b=1.25,a=0.0225,g=9.81,\gamma =7.25\times 10^{-5}$, $u_{*}$ is the friction velocity. $\Phi \left(k,\varphi \right)$ is the direction factor. Here, we use the spreading function given by Fung and Lee as
(17)$$\Phi \left(k,\varphi \right)=1+\frac{2}{1-N}\left(\frac{1-R}{1+R}\right)\left(1-\exp\left(-sk^{2}\right)\right)\cos\left(2\varphi \right)$$
(18)$$N=\frac{\int_{0}^{\infty }k^{2}s\left(k\right)\exp\left(-sk^{2}\right)dk}{\int_{0}^{\infty }k^{2}s\left(k\right)dk},\;R=\frac{0.003+0.00192U_{12.5}}{0.00316U_{12.5}}$$Here, $s=1.5\times 10^{-4}$, $U_{12.5}$ is the wind speed at 12.5 m above the sea level. Figure 9 shows two random rough sea surface samples generated by the Monte Carlo method and the DV spectrum when the wind speed is 5 and 10 m/s, respectively.

The manufacturing of simulated sea surface materials is based on molding the UV curing resin on the 3D printer workbench. Using the laser with specific intensity and wavelength, the surface of the light curing resin material is solidified in the order of point to line and line to surface. When the drawing of one layer is completed, the lifting table will move the height of one layer and then solidify the other layer [26]. In this way, layers are superimposed to form a three-dimensional solid. Figure 10a shows the geometry of the scaled sea surface model.

For the preparation of the dielectric coating on the rough sea surface, it is only necessary to spray equivalent dielectric coating on the surface of the local sea surface model. As a functional coating, the equivalent dielectric coating not only has certain commonalities but also has its own characteristics compared with ordinary coatings [27]. According to Section 3.4, the volume ratio of carbon black of the mixed material is optimized as 30%. After spraying a very thin thickness of mixed paint, the final scaled model is shown in Figure 10b,c. If the sample area is large and there are many coatings configured at one time, the disperser equipment shall be used to stir and disperse the configured coatings to prevent the long spraying process, which leads to the phenomenon of absorbent settlement in the configured coating.

## 4. Comparison of the RCS between the Prototype and Scaled Sea Surfaces

Firstly, the RCS of a single sea environment is simulated and analyzed to verify the sea surface based on the rough film medium meeting the scaling conditions under different conditions, and then the practical application of the scaled sea surface is verified by an RCS test of the scaled sea surface physical model.

### 4.1. RCS Simulation of the Rough Sea Surface

Suppose that the sea surface temperature of the prototype seawater is T=9.6 °C, the sea surface salinity is S=4 psu, and the electromagnetic wave frequency is f=7.8 GHz. The complex relative permittivity of the prototype sea surface is calculated as 61.3−36.2i according to the Debye model. With the above optimization algorithm, the volume ratio of carbon black is 28.5%, and the optimized complex relative permittivity is about 60.9−37.9i. Then, the polarization reflection coefficients of the prototype surface and the scaled surface are shown in Figure 11. The two sets of curves are adequately consistent.

Two wind speeds (2, 4 m/s) and two frequencies (7.75 GHz, 7.85 GHz) of the prototype rough sea surface are considered. The scaling coefficient is set as$\;K_{L}=3$, and the corresponding measurement frequencies of the scaled sea surface are 23.25 GHz and 23.55 GHz, respectively. The prototype sea surface is 1.5 × 1.5 m, and the scaled sea surface is 0.5 × 0.5 m. In order to eliminate the influence of rough surface edges as much as possible, the incident angle of the electromagnetic wave is set as 0~60°. Figure 12 shows the RCS of the prototype and scaled sea surfaces with different wind speeds. It can be seen that the RCS value of the sea surface decreases with the increase of wind speed. The curves of the RCS differences between the prototype sea surface and the scaled sea surface are stable for different wind speeds, frequencies, and incidence angles, and the average RCS difference is about 9.5 dB, which meets the RCS scaling theoretical formula in Equation (6).

### 4.2. Measurement of the Scaled Rough Sea Surface

In this section, the manufactured scaled sea surface in Figure 10 is used for measurement in the darkroom. The measurement system is shown in Figure 13, including instrument automatic control subsystem, turntable control subsystem, and RF measurement subsystem [28]. The RF subsystem mainly includes a vector network analyzer (VNA), power amplifier, directional coupler, and antenna. The vector network analyzer has a built-in signal source, and its RF output signal is amplified by the power amplifier and connected to the directional coupler. The signal at the coupling end of the directional coupler is connected to the reference port of the vector network analyzer as a reference signal. The direct signal of the coupler irradiates the target outward through the transmitting antenna, and the receiving antenna will collect the echo signal of the tested target and send it to the test port of the vector network analyzer. Finally, the amplitude and phase of the target echo is obtained by comparing the amplitude and phase of the test signal with the reference signal [29].

In order to measure the RCS of the scaled surface at different angles, the scaled rough surface needs to be placed on the turntable vertically; then the azimuth angle needs to be rotated by the turntable [30]. The test frequency is 23.4 GHz, and the selected calibrator is a metal plate of 0.18 m × 0.18 m. The low scattering support and background cancellation technology is used to effectively reduce the impact of background reflection such as the turntable, darkroom wall, antenna leakage, and target support on the test. The RCS of the test target at each frequency point and direction can be obtained according to Equation (10).
(19)$$\sigma_{T}=\sigma_{S}\times P_{T}/P_{S}$$Here, $\sigma_{T}$ is the RCS of the measured target and $P_{T}$ is the corresponding echo power. Similarly, $\sigma_{S}$ is the RCS of the metal plate calibrator, and $P_{S}$ is its echo power.

The scaled sea surface is 0.4 m × 0.4 m, the thickness can be obtained according to the scale factor, and the specific size is not constrained. The RCS test result curve of the scale sea surface is shown in Figure 14, compared with the RCS simulation result. With the change of the electromagnetic wave azimuth, the RCS curve of the scaled sea surface simulated and tested is relatively consistent. At each angle, the mean of the difference (absolute value) between the two curves is about 5.5 dB, which shows that the scaled sea surface model constructed in this paper has practical application.

## 5. Conclusions

This work proposes a scaled sea surface designing method based on a rough thin-film medium. Since the permittivity of sea water is dispersed according to the Debye mode, this paper discussed a mixed material with carbon black and epoxy, whose volume ratio is optimized with the genetic algorithm. This method can overcome the previous difficulty of adjusting the same permittivity of the water. The geometry of the scaled rough surface sample is numerically generated with the DV spectrum and is fabricated using 3D printing to keep the similar seawater shape. Then, the surface sample is sprayed with a layer of film material for the EM scattering measurement. The prototype sea surface and scaled sea surface models corresponding to different frequency and wind speeds are simulated to verify the scaling relationship between the scaled sea surface and prototype sea surface based on the rough film medium. Moreover, an indoor test system is constructed for RCS measurement of the scaled rough surface samples. The simulated and measured RCS (radar cross-section) results show good consistency for the prototype seawater and scaled materials, which indicates that the proposed scaled method is an efficient method to obtain the broadband EM scattering characteristics of rough sea surfaces.

## Figures and Tables

**Figure 1 sensors-22-06290-f001:**
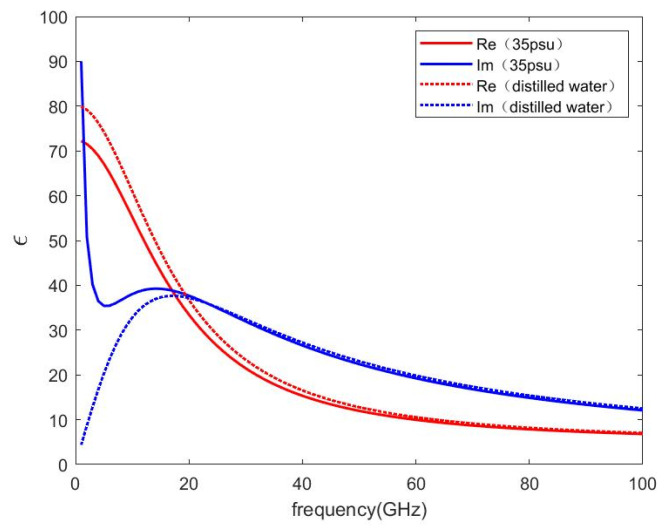
The permittivity of seawater varies with frequency. The red lines are for fresh water, and the blue lines are for salt water. The solid lines represent the real parts of the permittivity and the dotted lines represent imaginary parts. The water temperature T=20 °C. The salinity S=35 psu.

**Figure 2 sensors-22-06290-f002:**
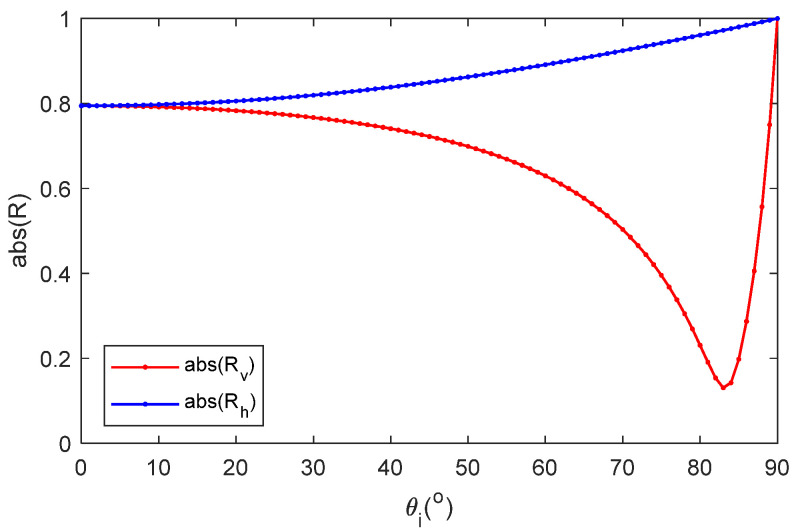
Reflection coefficient vs. incident angles for two polarizations. The blue line represents horizontal polarization. The red line represents vertical polarization (T=20 °C, S=30 psu, f=7.8 GHz).

**Figure 3 sensors-22-06290-f003:**
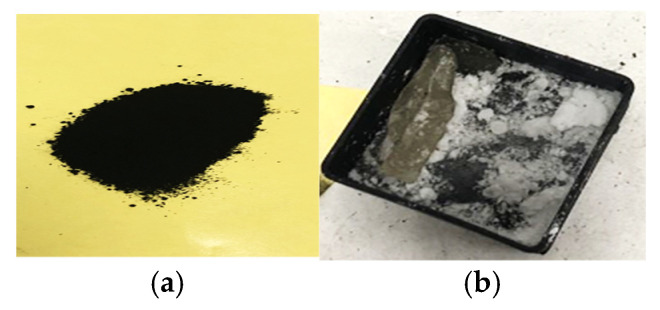
Material composition of the rough sea surface. The density of carbon black is 2.00 g/cm^3^ and the density of paraffin is 0.90 g/cm^3^. (**a**) Carbon black; (**b**) paraffin.

**Figure 4 sensors-22-06290-f004:**
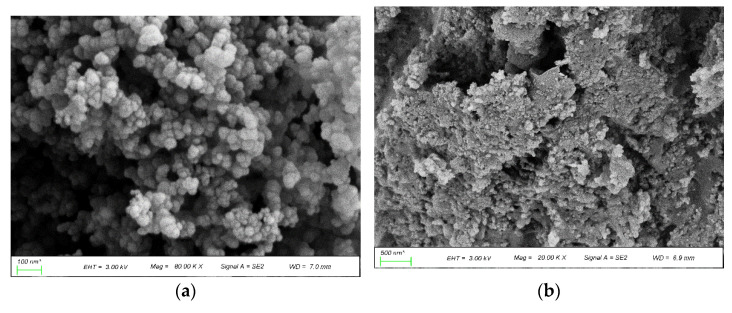
The morphology of the carbon black cluster and the particle dispersion in the resin. (**a**) Carbon black cluster; (**b**) carbon black cluster in the resin.

**Figure 5 sensors-22-06290-f005:**
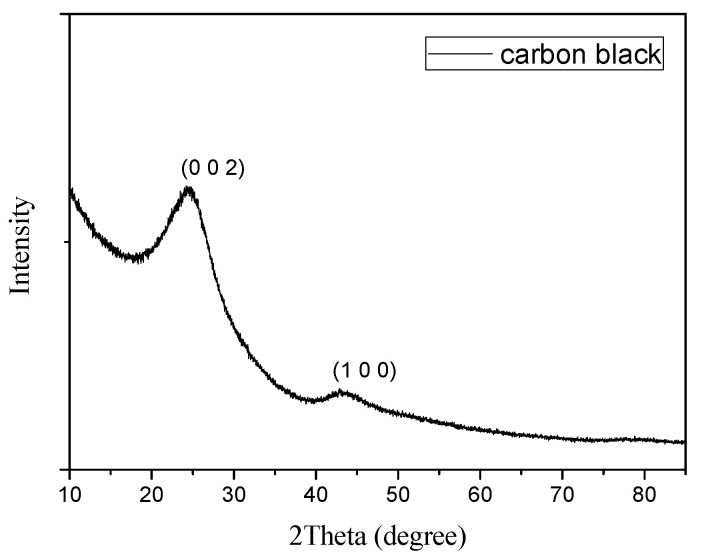
The XRD curve of carbon black.

**Figure 6 sensors-22-06290-f006:**
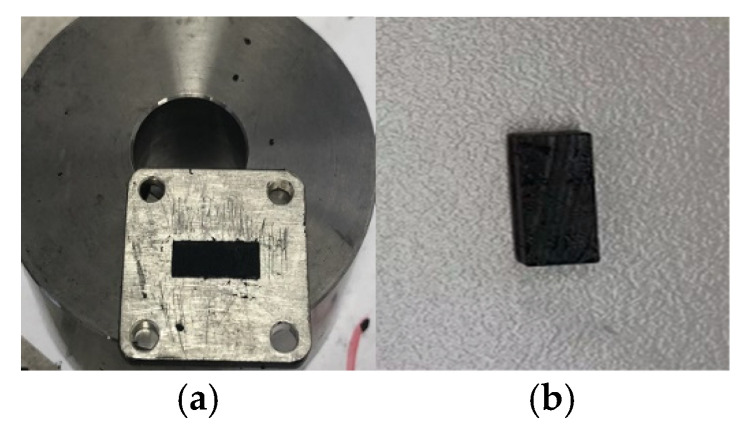
Preparation of mixed material samples. (**a**) Mixed material filled in the groove. (**b**) Rectangular sample for measurement.

**Figure 7 sensors-22-06290-f007:**
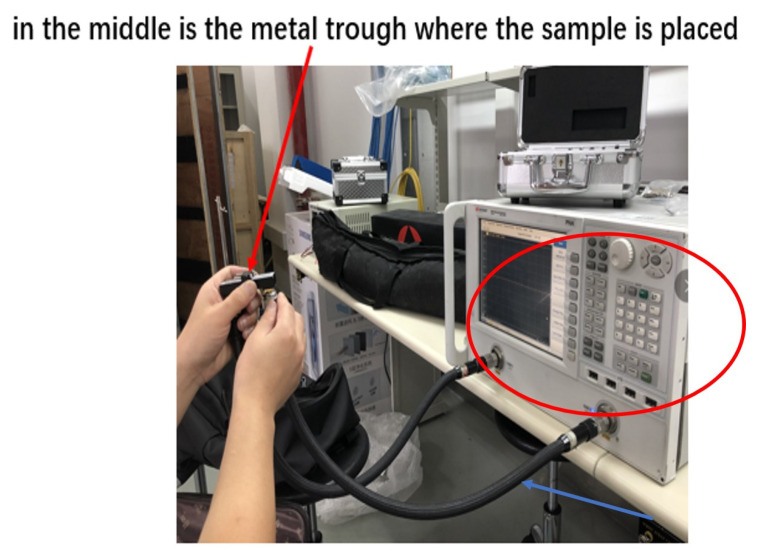
The transmission/reflection coefficient measurement system. Where the red arrow points is where the sample is placed. The instrument circled in red is a vector network analyzer (VNA). The blue arrow shows the coaxial line.

**Figure 8 sensors-22-06290-f008:**
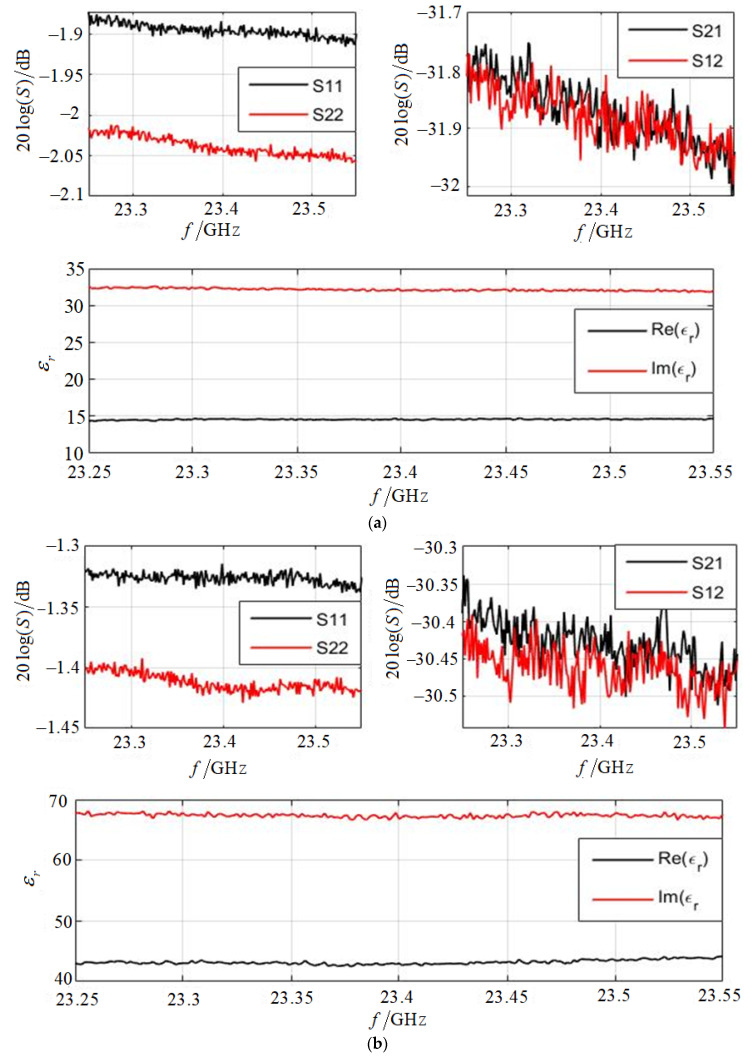
The measured scattering parameters and inversed permittivity. (**a**) Carbon black volume ratio is 25%; (**b**) carbon black volume ratio is 30%.

**Figure 9 sensors-22-06290-f009:**
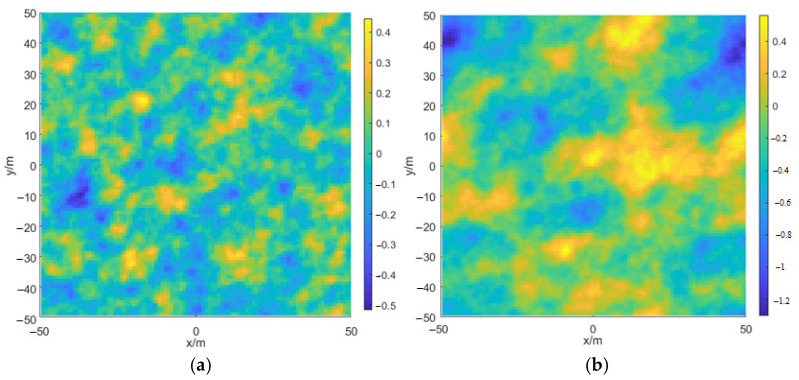
Rough sea surface sample with different wind speeds at 12.5 m above sea level. (**a**) $U_{12.5}=5\;m/s$; (**b**) $U_{12.5}=10\;m/s$.

**Figure 10 sensors-22-06290-f010:**
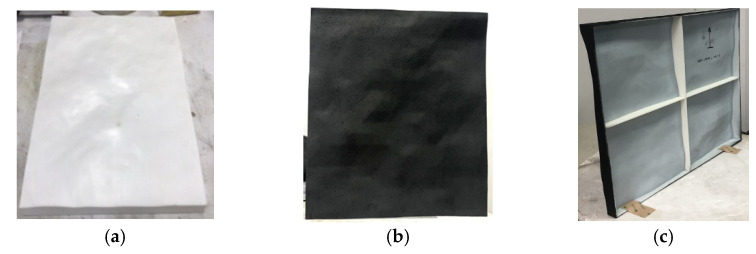
Scaled surface model for measurement. (**a**) Front view before spraying; (**b**) front view after spraying; (**c**) back view after spraying.

**Figure 11 sensors-22-06290-f011:**
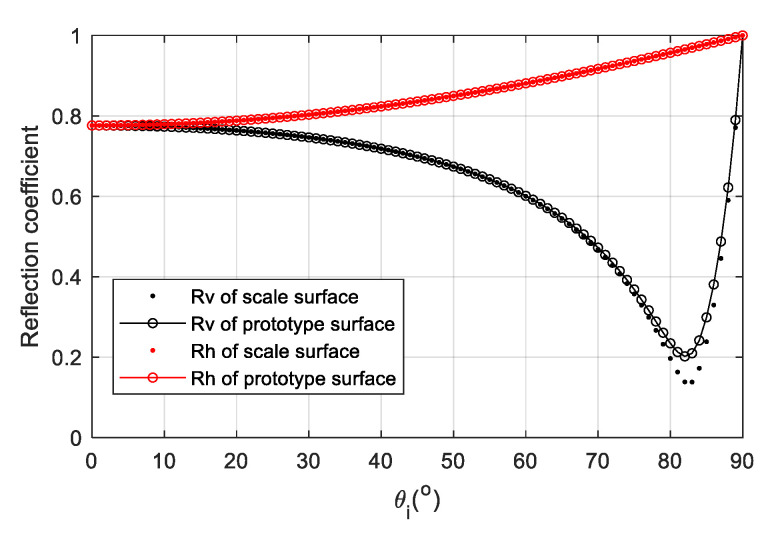
Reflection coefficient of the prototype surface and the scaled surface. The dot represents the reflection coefficient of the scaled sea surface. The solid line with the hollow circle represents the reflection coefficient of the prototype sea surface. The black represents vertical polarization. The red represents horizontal polarization (T=9.6 °C, S=4 psu, f=7.8 GHz).

**Figure 12 sensors-22-06290-f012:**
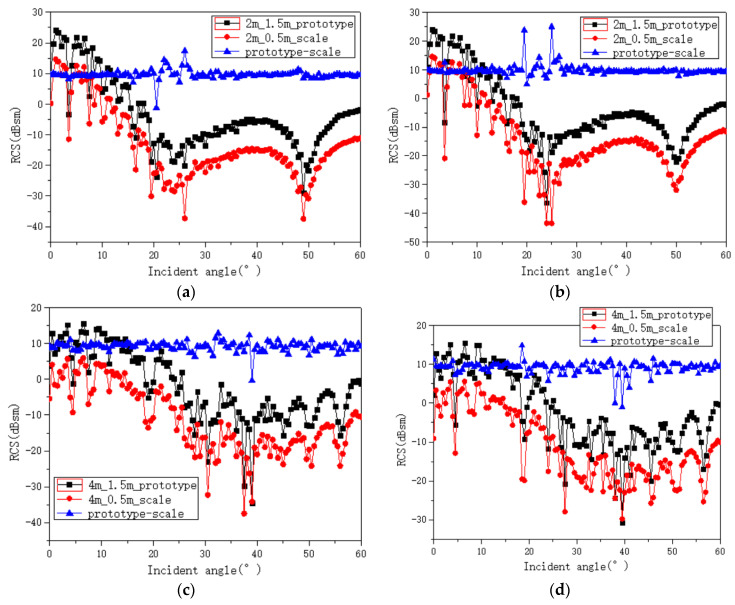
The RCS (dBsm) variation of the prototype and scaled rough sea surfaces with the incident angle. The black line represents 1.5 m × 1.5 m of the prototype sea surface. The red line represents 0.5 m × 0.5 m of the sea surface. The blue line shows the difference (T=9.6 °C, S=4 psu). (**a**) f=7.75 GHz, $U_{12.5}=2\;m/s$; (**b**) 7.85 GHz, $U_{12.5}=2\;m/s$; (**c**) f=7.75 GHz, $U_{12.5}=4\;m/s$; (**d**) f=7.85 GHz, $U_{12.5}=4\;m/s$.

**Figure 13 sensors-22-06290-f013:**
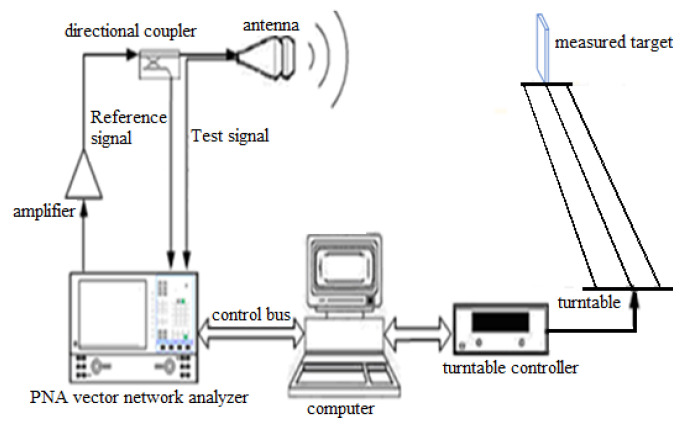
RCS test system in the electromagnetic darkroom.

**Figure 14 sensors-22-06290-f014:**
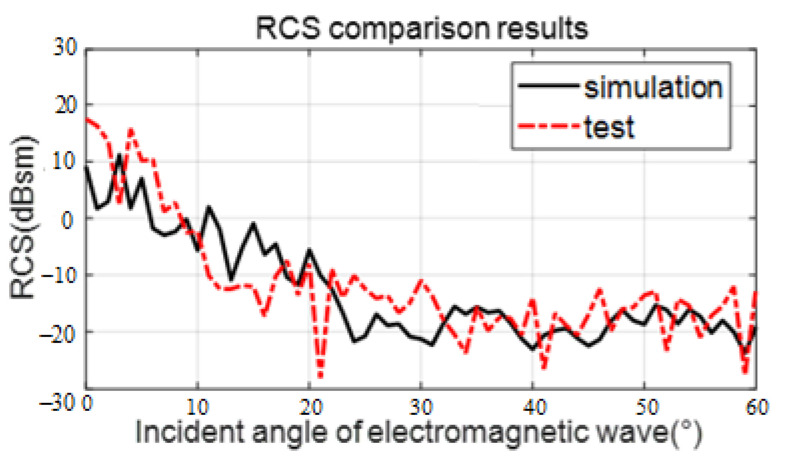
Comparison of scaled sea surface RCS between simulation and test. The solid black line represents the simulation results. The dashed red line represents the actual scaled model measurement results.

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
