# Peer review of "Scaled Sea Surface Design and RCS Measurement Based on Rough Film Medium"

_sensors, 2022, doi:10.3390/s22166290_

Round 1

Reviewer 1 Report

The contribution deals with fabrication of the sensors, which could simulate the surface of the see. This topic sound to me interesting. Nevertheless it is very difficult to read the paper and to understand the aim of the work and major revision has to be done before papers publication:

Abstract line 10-11: “The electromagnetic scattering characteristics of targets in rough sea environment is very important for target surveying and detection. The scaled measurement on the seawater becomes an efficient method to get the target characteristics.”  It will be good to specified which characteristic ?

Introduction:

The using RCS measuremens -  for readers will be useful to spell out the abbreviation RCS and give some citation, where one could find information about this topic.  I was reading this materials for example [IEEE Transactions on Antennas and PropagationVolume 62, Issue 2, Pages 945 - 949February 2014 Article number 6665095 and https://www.nsi-mi.com/images/Technical_Papers/2008/Introduction%20to%20RCS%20Measurements.pdf]

From the last paragraph for me is not obvious the aim of this study. Please specify more in detail.

Experimental:

The most important point is to fabricate the scaled materials by mixing the carbon black and paraffin. I am missing any information about the structure of carbon powder (grain size, structure performed for example by SEM, XRD, TEM etc.) and following structural characterisation of the samples (for example average distance between carbon grains). It must to be add, otherwise the readers don’t have basic information which follows on dielectric properties of the sample. Without this all informations are useless for other people and cannot be published in scientific journal.

Author Response

Dear reviewer:

Thank you for the helpful comments. We carefully revised the manuscript according to the reviewers’ opinions and replied to the comments one by one. All changes in the updated manuscript are indicated with the revision mode. The attached file is the one by one response.

Reviewer 2 Report

Guo and co-authors of the manuscript proposed a new method to design scaled sea surface using thin film material. Both simulated and measured results were presented with details. However, the introduction of the manuscript is unclear with little references were provided. Though the proposed method seems to be novel, it is still unclear how it can advance the existing methodologies as no real application was given. I highly suggest the authors include more details and motivations in the introduction, along with a figure showing the overall studies for the paper. Other comments can be found below:

1)      Please define RCS.

2)      Most of the statements in the introduction need reference.

3)      More discussions should be added in the conclusion.

4)      Figures might be merged and more details should be added to the figure captions.

Author Response

Dear reviewers:

Thank you for the helpful comments. We carefully revised the manuscript according to the review opinions and replied to the comments one by one. All changes in the updated manuscript are indicated with the revision mode. The attached file is the one by one response.

Round 2

Reviewer 1 Report

Congratulation to authors for good job. In my opinion after revisions the contribution is now much clear and will bring interesting information to science community.

Reviewer 2 Report

The authors have addressed my previous concerns.